# Cross-neutralizing antibodies bind a SARS-CoV-2 cryptic site and resist circulating variants

Tingting Li [1,2,9], Wenhui Xue[1,2,9], Qingbing Zheng [1,2,9], Shuo Song [3,4,9], Chuanlai Yang[1,2,9], Hualong Xiong[1,2,9], Sibo Zhang[1,2], Minqing Hong[1,2], Yali Zhang[1,2], Hai Yu[1,2], Yuyun Zhang[1,2], Hui Sun[1,2], Yang Huang[1,2], Tingting Deng[1,2], Xin Chi[1,2], Jinjin Li[1,2], Shaojuan Wang[1,2], Lizhi Zhou[1,2], Tingting Chen[1,2], Yingbin Wang[1,2], Tong Cheng [1,2], Tianying Zhang [1,2], Quan Yuan [1,2], Qinjian Zhao[1,2], Jun Zhang[1,2], Jason S. McLellan [5], Z. Hong Zhou [6,7✉], Zheng Zhang [3,4✉], Shaowei Li [1,2✉], Ying Gu [1,2✉] & Ningshao Xia [1,2,8✉]

The emergence of numerous variants of SARS-CoV-2, the causative agent of COVID-19, has presented new challenges to the global efforts to control the COVID-19 pandemic. Here, we obtain two cross-neutralizing antibodies (7D6 and 6D6) that target *Sarbecoviruses'* receptor-binding domain (RBD) with sub-picomolar affinities and potently neutralize authentic SARS-CoV-2. Crystal structures show that both antibodies bind a cryptic site different from that recognized by existing antibodies and highly conserved across *Sarbecovirus* isolates. Binding of these two antibodies to the RBD clashes with the adjacent N-terminal domain and disrupts the viral spike. Both antibodies confer good resistance to mutations in the currently circulating SARS-CoV-2 variants. Thus, our results have direct relevance to public health as options for passive antibody therapeutics and even active prophylactics. They can also inform the design of pan-sarbecovirus vaccines.

[1] State Key Laboratory of Molecular Vaccinology and Molecular Diagnostics, School of Life Sciences, School of Public Health, Xiamen University, 361102 Xiamen, Fujian, China. [2] National Institute of Diagnostics and Vaccine Development in Infectious Diseases, Xiamen University, 361102 Xiamen, Fujian, China. [3] Institute for Hepatology, National Clinical Research Center for Infectious Disease, Shenzhen Third People's Hospital, 518112 Shenzhen, Guangdong, China. [4] The Second Affiliated Hospital, School of Medicine, Southern University of Science and Technology, 518112 Shenzhen, Guangdong, China. [5] Department of Molecular Biosciences, The University of Texas at Austin, Austin 78712 TX, USA. [6] California NanoSystems Institute (CNSI), UCLA, Los Angeles 90095 CA, USA. [7] Department of Microbiology, Immunology and Molecular Genetics, University of California, Los Angeles, Los Angeles 90095 CA, USA. [8] Research Unit of Frontier Technology of Structural Vaccinology, Chinese Academy of Medical Sciences, 518112 Xiamen, Fujian, China. [9] These authors contributed equally: Tingting Li, Wenhui Xue, Qingbing Zheng, Shuo Song, Chuanlai Yang, Hualong Xiong. ✉email: hong.zhou@ucla.edu; zhangzheng1975@aliyun.com; shaowei@xmu.edu.cn; guying@xmu.edu.cn; nsxia@xmu.edu.cn

S evere acute respiratory syndrome coronavirus 2 (SARS-CoV-2), responsible for the ongoing global coronavirus disease 2019 (COVID-19) pandemic, is the third human coronavirus to cause widespread infection[1]. It has already claimed nearly 4.01 million lives and contributed to over 185.2 million confirmed cases in 219 countries and territories as of 8 July 2021, with broad consequences for public health and global economy[1–3]. Antibodies from convalescent donors have shown efficacy in treating COVID-19 caused by the original virus strain of SARS-CoV-2 (refs. [4,5]), and antibody cocktails have been clinically prescribed under Emergency Use Authorization (EUA) to patients, as in a well-publicized case[6]. More recently, several vaccines that confer excellent COVID-19 protection efficacy have been administered broadly under EUA or (in some countries) full approval[7]. However, the emerging SARS-CoV-2 variants escape neutralization by some of the more potent neutralizing antibodies, diminish the effectiveness of the approved vaccines, and reduce the efficacy of existing antibody cocktail treatment[8–10]. For example, the Moderna (mRNA-1273), Pfizer (BNT162b2), and Novavax (NVX-CoV2373) vaccines show lower protection efficacy for these epidemic variants than the original strains[2,11]. A more broadly protective vaccine that could prevent infection of known and future variants would clearly be preferable, and broadly neutralizing antibodies could be used as both therapeutics and prophylactics[12,13]. Such antibodies can also be used to identify highly conserved antigenic determinants across various coronavirus strains to guide design of a broad-spectrum vaccine and to serve as indicators for cross-protection potential upon vaccine immunization.

Transmission between individuals is driven by the spike glycoprotein (S) on the viral surface[14,15]. Each S monomer is proteolytically cleaved into S1 and S2 subunits prior to mediating virus entry into host cells by interacting with surface receptors[14,15]. Both SARS-CoV-2 and SARS-CoV enter host cells by engaging with the angiotensin-converting enzyme 2 (ACE2) receptor, which is recognized by the receptor-binding domain (RBD) of S[15]. To bind ACE2, the RBD must shift from a closed to an open conformation, and subsequent shedding of S1 allows the large-scale fusogenic transition of S2 (refs. [14,16,17]). Therefore, S protein and host factors related to cell entry of SARS-CoV-2 are promising targets for therapeutics development against COVID-19 (refs. [18–20]). Among reported potently neutralizing antibodies (nAbs), most neutralize coronaviruses by targeting the receptor-binding motif (RBM) within the RBD to block S from engaging with ACE2 (refs. [5,21,22]). However, many of these nAbs are specific to either SARS-CoV-2 or SARS-CoV, and some do not neutralize variant strains[8]. To date, at least two classes of cross-neutralizing antibodies have been identified, represented by CR3022 (refs. [23,24]) and S309 (ref. [25]). These nAbs recognize two non-overlapping conserved epitopes distal from the RBM. The neutralization activity of S309 was found to be impaired by mutations in the B.1.1.7 (Alpha) variant[8]; the neutralizing activity of CR3022 for that variant remains to be determined.

In this study, we raised two cross-neutralizing antibodies, 7D6 and 6D6, by immunization of mice with the SARS-CoV-2 S-trimer alone (7D6) or in combination with the SARS-CoV S-trimer and MERS-CoV RBD (6D6). High-resolution crystal structures reveal that both antibodies target a distinctive cryptic site of the RBD with high conservation. Our results thus help expand the epitope coverage for antibody cocktail therapies for emerging variants, and augment existing strategies for designing pan-sarbecovirus vaccines.

## Results

### Cross-neutralizing antibodies elicited by combined immunization of coronavirus spikes. To obtain cross-neutralizing antibodies, we explored immune response to combined immunization of *Sarbecoviruses*. Mice were immunized with

SARS-CoV-2 spike protein alone and in combination with the SARS-CoV spike protein and the MERS-CoV RBD[26] (Fig. 1a, b). The hybridoma cell pools with reactivities against both SARS-CoV-2 and SARS-CoV spikes were selected and screened for cross-reactive monoclonal antibodies (mAbs). We obtained 5 and 10 lead mAbs from the single and combined immunization strategies, respectively (Fig. 1c and Table S1). Four of the five mAbs derived from the SARS-CoV-2 spike protein immunization recognized the S2 protein, whereas the fifth mAb targeted the RBD. From the combined immunization, only one of ten antibodies recognized the S2 protein, and the remaining nine mAbs recognized S1, eight of which targeted the RBD.

We next evaluated the neutralization potency of these 15 mAbs using the lentiviral virus (LV)[27] and vesicular stomatitis virus (VSV) pseudotyping systems[28]. A total of seven mAbs—one from the SARS-CoV-2 spike protein immunization and six from the *Sarbecovirus* spike immunization—showed cross-neutralizing activities against SARS-CoV-2 and SARS-CoV. All seven cross-neutralizing mAbs recognized the RBD (Table S1). In terms of the overall profiles of cross-reactivity and cross-neutralization, three mAbs—7D6, 6D6, and 16D8—stood out as lead antibodies and conferred potent cross-neutralization, with IC$_{50}$ value ranging from 2.56 to 8.91 µg/mL for SARS-CoV-2 pseudotyped LV (LV-SARS-CoV-2), 1.21 to 10.11 µg /mL for LV-SARS-CoV, and 0.04 to 0.26 µg/mL for VSV-SARS-CoV-2 (Fig. 2a–c). By contrast, the cross-neutralizing antibody CR3022 had poor neutralizing activity against LV-SARS-CoV-2 (IC$_{50}$ > 284 µg/mL) in our assay (Fig. 2a).

**Characterization of cross-neutralizing antibodies**. We selected the three promising mAbs for further investigation—7D6 (raised by single immunization strategy), and 6D6 and 16D8 (raised by combined strategy)—and evaluated their neutralization activities against authentic SARS-CoV-2 (ref. [5]). All three cross-neutralizing mAbs neutralized SARS-CoV-2 with comparable IC$_{50}$ values: 2.23, 1.77, and 5.30 µg/mL, respectively (Fig. 2d). Using surface plasmon resonance (SPR) analysis, we found that these neutralizing mAbs interacted with the two-proline stabilized S trimer (S-2P) and RBD with nanomolar or subnanomolar affinities, for both SARS-CoV and SARS-CoV-2 strains (Fig. 2e, Fig. S2 and Table S2). In particular, 7D6 bound the RBD with a 2-log higher affinity than the trimeric S-2P, and 6D6, which showed comparable binding affinities to the two proteins, had 2-log higher affinities for SARS-CoV-2 proteins than SARS-CoV proteins (Fig. 2e). 16D8 showed the same RBD-binding preference as did 7D6, but like 6D6, a lower affinity for SARS-CoV proteins than for SARS-CoV-2 proteins. A blocking assay confirmed that 16D8 inhibited binding of ACE2 with the RBD, while 7D6 and 6D6 did not (Fig. 2f). In a competition assay, we found that 6D6 completely blocked 7D6 from binding the RBD (Fig. 2g) and that 16D8 could block neither 7D6 nor 6D6. These results indicate overlap of the 7D6 and 6D6 epitopes and that these epitopes are distinct from the RBM and from the binding sites of most other RBD-directed mAbs as well as from the epitope of 16D8, which probably overlaps the RBM.

**Structural basis of 7D6/6D6 cross-neutralization**. We characterized the details of binding between these antibodies and SARS-CoV-2 spikes by determining crystal structures of Fab-RBD immune complexes. We obtained crystals of SARS-CoV-2 RBD bound with Fabs of 7D6, 6D6, or 16D8 and determined the structures at 1.40 and 1.92 Å resolutions for RBD:7D6 and RBD:6D6 complexes, respectively (Fig. S1 and Table S3). We showed that 7D6 and 6D6 bind, in similar orientations, to a nearly identical region of the RBD (Fig. 3a, d). Their common

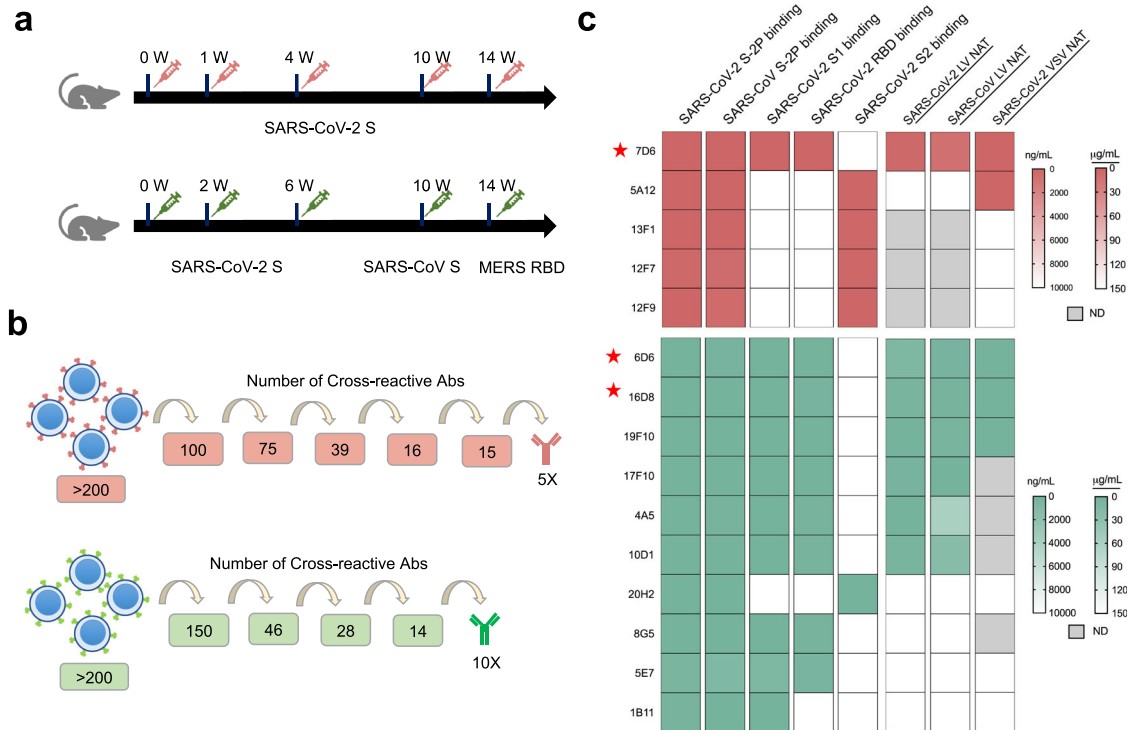

**Fig. 1 Screening and characterization of SARS-CoV-2 and SARS-CoV cross-reactive monoclonal antibodies. a** The immunization schemes of two different strategies: immunized with the SARS-CoV-2 spike protein alone (upper) or with a combination of the SARS-CoV-2 and SARS-CoV spike proteins and the MERS-CoV RBD (lower). **b** The number of lead antibodies with cross-activity in each round of screening. **c** Characterization of 5 and 10 monoclonal antibodies obtained by the first and second immunization strategies, respectively. The neutralizing titers ($IC_{50}$) based on the vesicular stomatitis virus pseudotyping system (VSV NAT) and the lentiviral virus pseudotyping system (LV NAT) were tested. ND means not detected. Asterisks indicate the top three monoclonal antibodies with excellent cross-neutralizing potency.

epitope is distal to the RBM (Fig. 3g), consistent with the SPR results above (Fig. 2f). The paratopes of 7D6 and 6D6 comprise six and five complementarity determining region (CDR) loops, respectively, with a buried area at the antibody-RBD interface of about 796 Å$^2$ (HCDRs accounting for 73.3%; LCDRs, 26.7%) and 1,019.1 Å$^2$ (HCDRs, 54.7%; LCDRs, 45.3%), respectively (Fig. 3b, e). The epitopes for 7D6 and 6D6 are formed by 21 and 25 residues, respectively, most of which are in the η1, β1, β5, and β7–β8 RBD loops, with 18 shared residues (Fig. 3c, f, i and Fig. S2). Their contacts are conserved in the SARS-CoV-2 variants: 21 (21/21, 100%) conserved residues for binding with 7D6 and 25 (25/25, 100%) conserved residues for 6D6 (Fig. 3i). The 18 shared residues within the epitope are conserved (89.0%) across *Sarbecovirus* isolates including clade 1, 2, 3 (ref. [29]) (Fig. 3h and Table S4); we denote this shared region as the 7D6/6D6 site.

7D6 interacts with the SARS-CoV-2 RBD mainly through eight hydrogen bonds and van der Waals contacts between HCDR1-3 and residues 346–355 and 466–471 of the RBD (Fig. 3j); RBD residues T470, Y351, R346, R466, and R355 are critical for these interactions (Fig. 3k). For 6D6, the binding site comprises predominantly two RBD segments, one from residues 351 through 357 and the other from residues 457 through 471 (Fig. 3l). A strong and extensive interaction network, including 15 hydrogen bonds and several van der Waals contacts, is formed between 6D6 and the RBD, with T470, E471, Y351, R466, F464, R355, P463, R457, N460, E465, and R357 acting as key residues in the antigenic determinant (Fig. 3m). Among the critical residues in the 7D6 and 6D6 epitopes, there are five amino acid variations between SARS-CoV-2 and SARS-CoV (T470 in SARS-CoV-2 to N in SARS-CoV, R346→K, E471→V, N460→K, R357→K), but these differences do not substantially affect cross-reactivity (Fig. 2a, b).

**Mutation resistance of 7D6/6D6 and classification of SARS-CoV-2 RBD nAbs.** Over the past year, antibody resistance has developed in SARS-CoV-2 variants, including B.1.1.7 (Alpha) and B.1.351 (Beta)[8]. Mutations in the RBM include N501Y in the B.1.17 variant, K417N/E484K/N501Y in the B.1.351 variant, K417T/E484K/N501Y in the P.1 (Gamma) variant, L452R/T478K in the B.1.617.2 (Delta) variant, L452R/E484Q in the B.1.617.1 (Kappa) variant, E484K in the B.1.526 (Iota) variant, and L452Q/F490S in the C.37 (Lambda) variant[30]. All these positions are outside the 7D6/6D6 site (Fig. 4a), and 7D6 and 6D6 maintain, as expected, strong reactivities for the RBD in proteins bearing these typical mutations (Fig. 4b). In contrast, COVID-19 convalescent sera had lower reactivities to RBD mutants, although to varying extents. Moreover, 7D6 and 6D6 showed nearly unchanged neutralizing potency against LV-SARS-CoV-2 of the circulating B.1.1.7, B.1.351, P.1 variants, and as well as against the B.1.351 authentic virus. Control REGN10933 had significantly lower activity against LV-SARS-CoV-2 of B.1.351 and P.1, and unchanged activity against B.1.1.7 (Fig. 4b and Figs. S3–S4).

A global competition analysis of RBD antibodies[31] has allowed the RBD-targeting nAbs to be grouped into three classes (Class 1–3) (Fig. 4c). Class 1 includes antibody S309 (ref. [25]), which has a footprint distinct from the RBM and from the CR3022 epitope. The 7D6/6D6 site is also away from the RBM, proximal to the S09 epitope but non-overlapping. We therefore group 7D6 and 6D6 into Class 1, and suggest that they may have properties distinct from those of S309 because they appear to destabilize the trimeric spike (Fig. 4c, d and Table S5). Antibodies in Class 2 are SARS-CoV-2 specific, as represented by antibody CB6 and others[4,5,21,22,32–51]. The epitopes for antibodies in Class 2 overlap to varying extent with the ACE2 binding site and likely neutralize

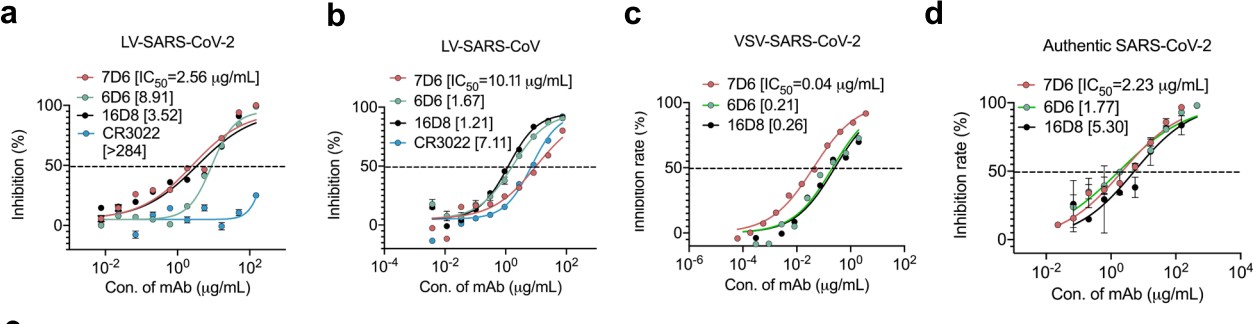

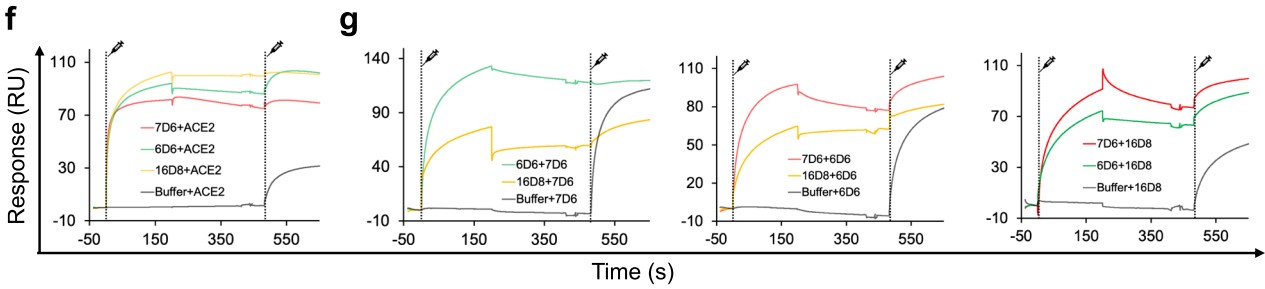

**Fig. 2 Comprehensive characterization of representative neutralizing mAbs 7D6, 6D6, and 16D8. a–c** Neutralization of three mAbs by LV-SARS-CoV-2 (**a**), LV-SARS-CoV (**b**), and VSV-SARS-CoV-2 (**c**). **d** Neutralization activities of three mAbs to authentic SARS-CoV-2. **e** SPR kinetics and the affinities of three antibodies to S-2P and RBD proteins of both SARS-CoV-2 and SARS-CoV. **f** SPR-based blocking assays of the three mAbs perturbing the engagement of ACE2 to RBD. **g** Inter-blocking potentials of the three mAbs. Data in **a**, **b**, **d** are presented as mean values ± SEM. Source data are provided as a Source Data file.

the virus by blocking ACE2 from binding to the RBD. Class 3 has a total of eight antibodies including four cross-neutralizing ones[23,39,51–55]; CR3022, which falls in this class, targets a cryptic epitope on the outer side of the RBD and may cause neutralization by antibody-induced spike disruption[24]. Antibodies in Classes 1 and 3 bind two separate regions distal to the RBM. For COVID-19 therapy, passive treatment with individual antibodies or with non-competing antibody cocktails is an effective approach; three antibodies or combinations have been approved under Emergency Use Authorization (EUA) (Fig. 4e), although these have somewhat lower efficacy for the emerging variants[8–10]. Thus, 7D6 and 6D6 bind a conserved site distinct from these other clinical antibodies (Fig. 4e), and their appropriate humanized forms could be potential candidate therapeutics, or enhancers to existing cocktails, to protect against variants.

**Mechanism of 7D6/6D6-mediated neutralization.** We docked the two immune-complex structures into the structure of the trimeric spike, superposing the RBD in either its open or closed conformation. In all interaction scenarios, the bound Fabs would clash with the adjacent NTDs (Fig. 5a, b, e, f, i, j, m, n). The overlapping volumes of 7D6 to the adjacent NTD in the models were 3,700 Å³ for RBD up and 11,800 Å³ for RBD down, and the occluded volumes for 6D6 were 3,700 Å³ (RBD up) and 13,100 Å³ (RBD down) (Fig. 5c, g, k, o). Therefore, the 7D6/6D6 site, located

on the inner aspect of the RBD facing the adjacent NTD, is cryptic in the context of the entire trimeric spike (Fig. 5d, h, l, p).

We then examined binding of 7D6 or 6D6 to S-2P in vitro. High-performance liquid chromatography (HPLC) analysis showed that S-2P incubated with 7D6 or 6D6 Fab dissociates into smaller components (Fig. 6a, b). Cryo-electron microscopy imaging of S-2P in complex with 7D6 or 6D6 Fab and 2D classification analyses of the resulting complexes confirmed antibody-mediated disruption of the trimeric spike (Fig. 6c). We then carried out the shedding assay using the full-length wild type S (S-WT) expressed on the surface of 293T cells and quantified the results by flow cytometry. Both full-length and Fab forms of antibodies 7D6 and 6D6 triggered spike shedding of up to 63% after incubating with cells for 120 min, more potently than control CR3022 (Fig. 6d). These results confirm our proposed neutralization mechanism for 7D6/6D6 by antibody-induced spike shedding (Fig. 6e).

## Discussion

The mechanism of neutralization by the SARS-CoV-2 cross-neutralizing antibody CR3022 is by antibody-induced spike disruption[24]. We likewise showed previously that the 8C11 antibody can neutralize native hepatitis E virus (HEV) by antibody-imposed physical disruption[56]. Although the 7D6/6D6 site is inaccessible for antibody binding to the trimeric spike, 7D6 and 6D6 still associate with the S-2P protein with good

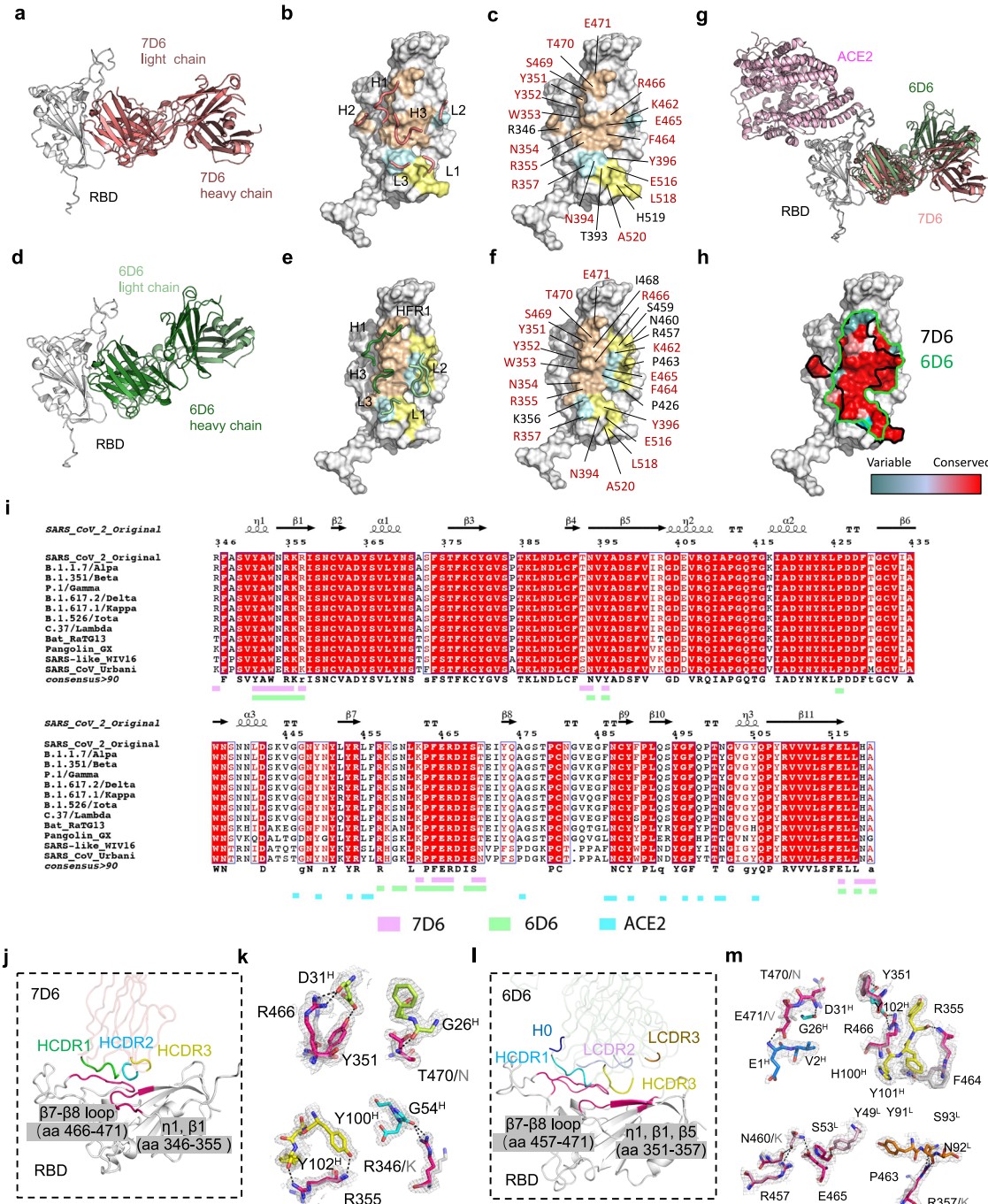

**Fig. 3 Crystal structures of 7D6 and 6D6 in complex with SARS-CoV-2 RBD. a**, **d** Overall structures of 7D6:RBD and 6D6:RBD. **b**, **e** CDRs of 7D6 (**b**) and 6D6 (**e**) involved in the interactions. RBDs and CDRs are shown as surface and cartoon, respectively. The contact surfaces of the heavy and light chains on the RBD are colored in orange and yellow, respectively; residues contacting both the heavy and light chains are colored in cyan. **c**, **f** Residues on the RBD involved in the interactions with 7D6 (**c**) and 6D6 (**f**) are indicated; shared residues are labeled in red. **g** Superimposition of the ACE2:RBD complex (PDB: 6M0J) and our immune complexes, revealing no competition between ACE2 and the mAbs. **h** Conservation analysis of critical residues in the binding epitopes for 7D6 (outlined in black) and 6D6 (outlined in green). SARS-CoV-2 (N = 2,216,094) and other SARS-related (N = 83) genes encoding for Sarbecovirus spike proteins were selected to calculate conservation. Deeper red indicates more conservation. **i** Sequence alignment of the RBDs of various Sarbecoviruses. Secondary structure distribution is indicated according to the RBD structure in 7D6:RBD co-crystal structure (PDB no. 7EAM). The plot was prepared by the online server (https://espript.ibcp.fr/ESPript/cgi- bin/ESPript.cgi). The footprints of ACE2, 7D6, and 6D6 are marked in cyan, pink, and green, respectively. **j** Interaction between the 7D6 heavy chain variable region and the RBD. The contact region on the RBD is colored purple, and HCDR1-3 are colored in green, cyan, and yellow, respectively. **k** Hydrogen bonds (dashed lines) between 7D6 and RBD. The electron density (2Fo−Fc) map of all residues is displayed on the contour level of 1σ above the mean value. Conserved residues between SARS-CoV-2 and SARS-CoV are shown in stick mode and in the same color scheme as in (**j**) for SARS-CoV-2 and in gray for SARS-CoV. **l** Interactions between the 6D6 CDRs and RBD. The contact region on the RBD is colored in purple; H0, HCDR2, HCDR3, LCDR2, and LCDR3 are colored in dark blue, cyan, yellow, pink, and orange, respectively. **m** Hydrogen bonds (dashed lines) between 6D6 and RBD.

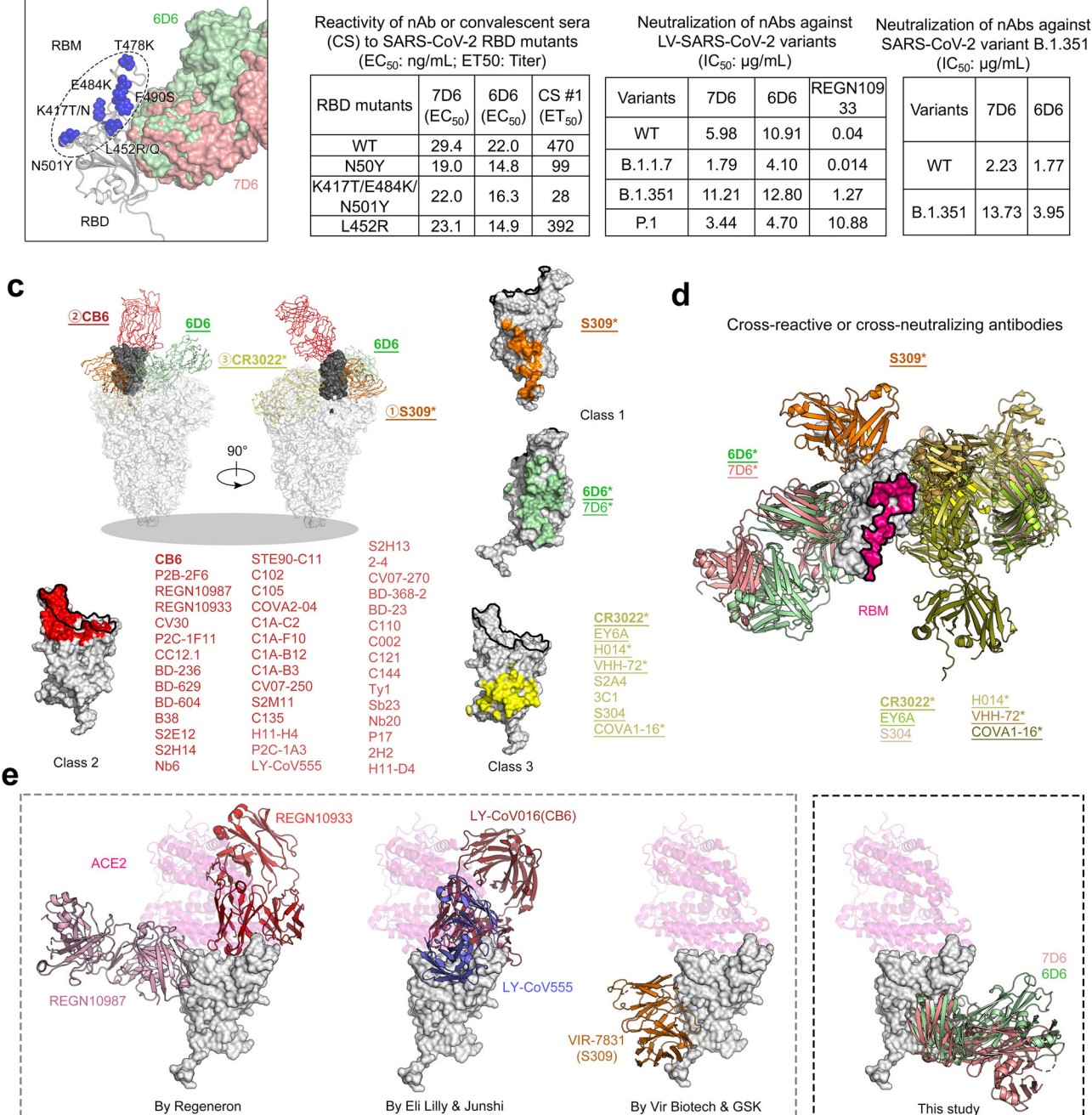

**Fig. 4 Mutation resistance of 7D6 and 6D6 and comparison of the 7D6/6D6 site with other binding modes of known RBD nAbs. a** Predominant mutations on the RBD in the SARS-CoV-2 variants. None of these mutations occur within the 7D6/6D6 site. **b** Binding reactivities of 7D6, 6D6 against the RBD mutants and their neutralizing activities against pseudotyped LVs and authentic virus of the major SARS-CoV-2 variant(s). Three convalescent sera and nAb REGE10933 served as control. **c** Superimposed structures of S trimers (shown as gray surface) and three classes of nAb-RBD complexes, the bound RBD were highlighted in dark gray. The epitopes of the six representative nAbs were shown on RBD separately. nAbs belonging to the same class (with similar epitopes or binding modes) are listed in the corresponding color. The underlined antibodies indicate cross-reactive antibodies and the asterisks indicate cross-neutralizing antibodies. **d** Superimposed structures of the Class 3 cross-reactive nAbs (CR3022, Ey6A, S304, H014, VHH-72, and COVA1-16), Class 1 cross-reactive nAb (S309, 7D6, and 6D6), showing three different binding orientations. Both two classes of nAbs bind epitopes without overlapping the RBM. **e** Comparison of 7D6, 6D6 and antibodies clinically used under EUA. REGN10933 + REGN10987 developed by Regeneron Pharmaceuticals Inc., LY-CoV16 (CB6) + LY-CoV555 from Eli Lilly & Company and Junshi Biosciences Inc., VIR-7831(S309) from Vir Biotechnology and GlaxoSmithKline group of companies (GSK). Source data are provided as a Source Data file.

affinities (Fig. 2e), suggesting that the RBD might have a range of orientations with respect to the NTD, allowing the antibody to bind upon transient exposure of the 7D6/6D6 site. That is, the cryptic 7D6/6D6 site may be exposed transiently by inter-domain

movement, as found for the hemagglutinin of the influenza virus[57,58]. Despite the similar binding orientations and footprints of the two antibodies, 7D6 has higher affinity for the RBD than for S2P, while the corresponding affinities for 6D6 are

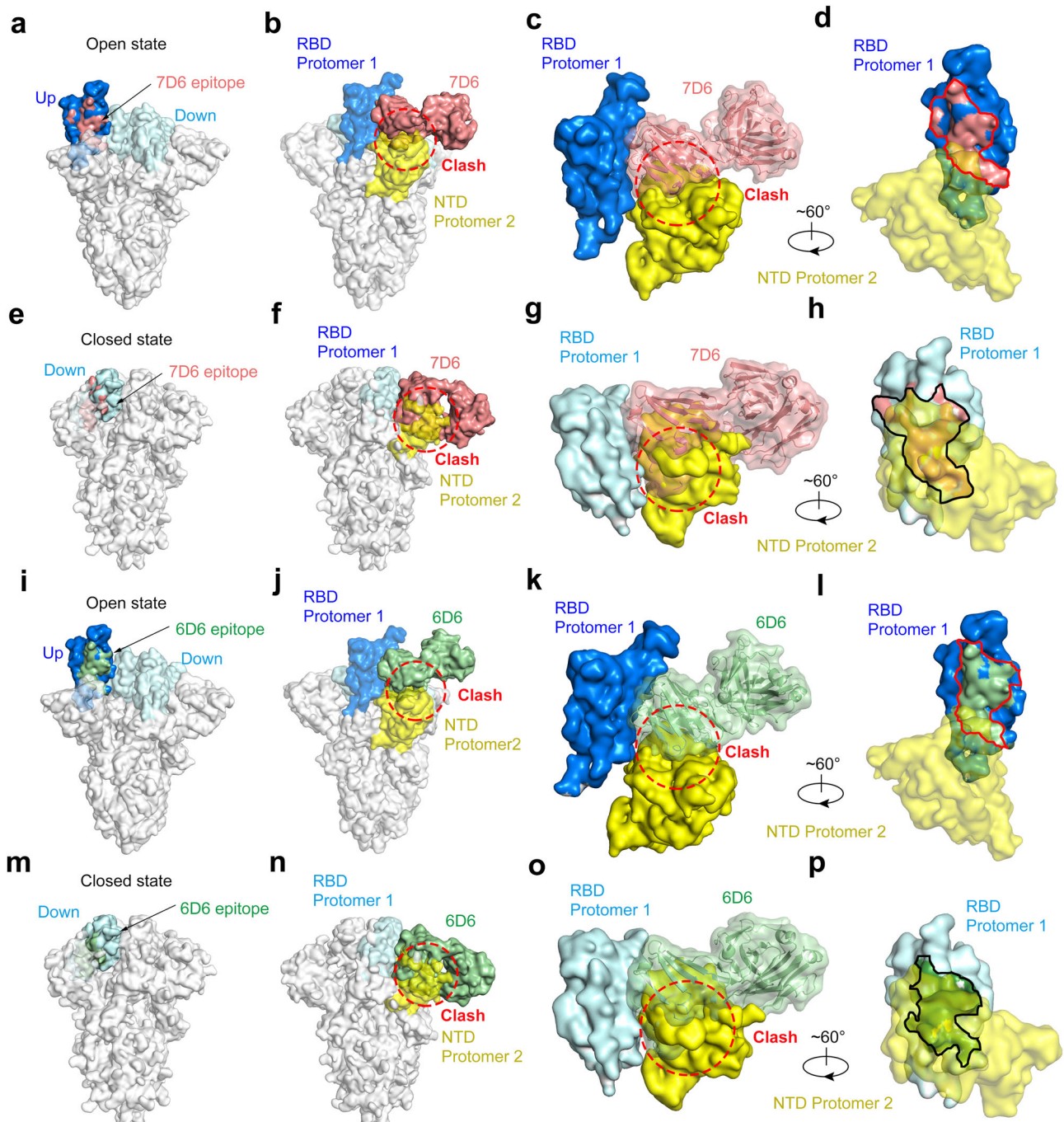

**Fig. 5 Structural analyses reveal the spatial clash induced by 7D6 and 6D6 binding. a**, **e**, **i**, **m** Epitope locations of 7D6 and 6D6 on the trimeric spike (PDB: 6zgg) when the RBD is in its open (blue) or closed (cyan) states (PDB:6vxx). 7D6 are colored in salmon and 6D6 in pale green. **b**, **f**, **j**, **n** Superimposition of the 7D6:RBD or 6D6:RBD complex and the open or closed RBD on the trimeric spike. Binding of 7D6 or 6D6 to either the open or closed state of the RBD would cause clashing (indicated by the red dash circles) with the neighboring NTD. **c**, **g**, **k**, **o** A close-up view of the clash of 7D6 and 6D6 on RBD. **d**, **h**, **l**, **p** Cryptic feature of the epitopes of 7D6 and 6D6 is shown on RBD in surface mode with a neighboring NTD rendered as semi-transparent surface, with close-up views of (**b**), (**f**), (**j**), (**n**), respectively.

comparable (Fig. 2e), perhaps because the epitope of 7D6 is more inaccessible in the structural context of S trimer than that of 6D6 (Fig. 5c, g, k, o). We propose 7D6/6D6 might neutralize virus by S1 shedding (Fig. 6e), as supported by antibody-induced spike shedding in vitro (Fig. 6d).

Several COVID-19 vaccines are already available and in use, and over 200 candidates are under development, mostly using the spike protein sequence from the original SARS-CoV-2 strain[7]. This study demonstrates that combined immunization with

multiple *Sarbecoviruses* spikes can produce more cross-neutralizing antibodies than immunization with SARS-CoV-2 spike only. Vaccination regimens with a combined and/or sequential immunization strategy might provide cross-immunity for SARS-CoV-2 and SARS-CoV as well as for circulating variants by focusing the response on cross-neutralizing epitopes such as the conserved 7D6/6D6 site. The potential for affinity maturation in the development of human B cell memory during long-term immunization might be part of such an approach.

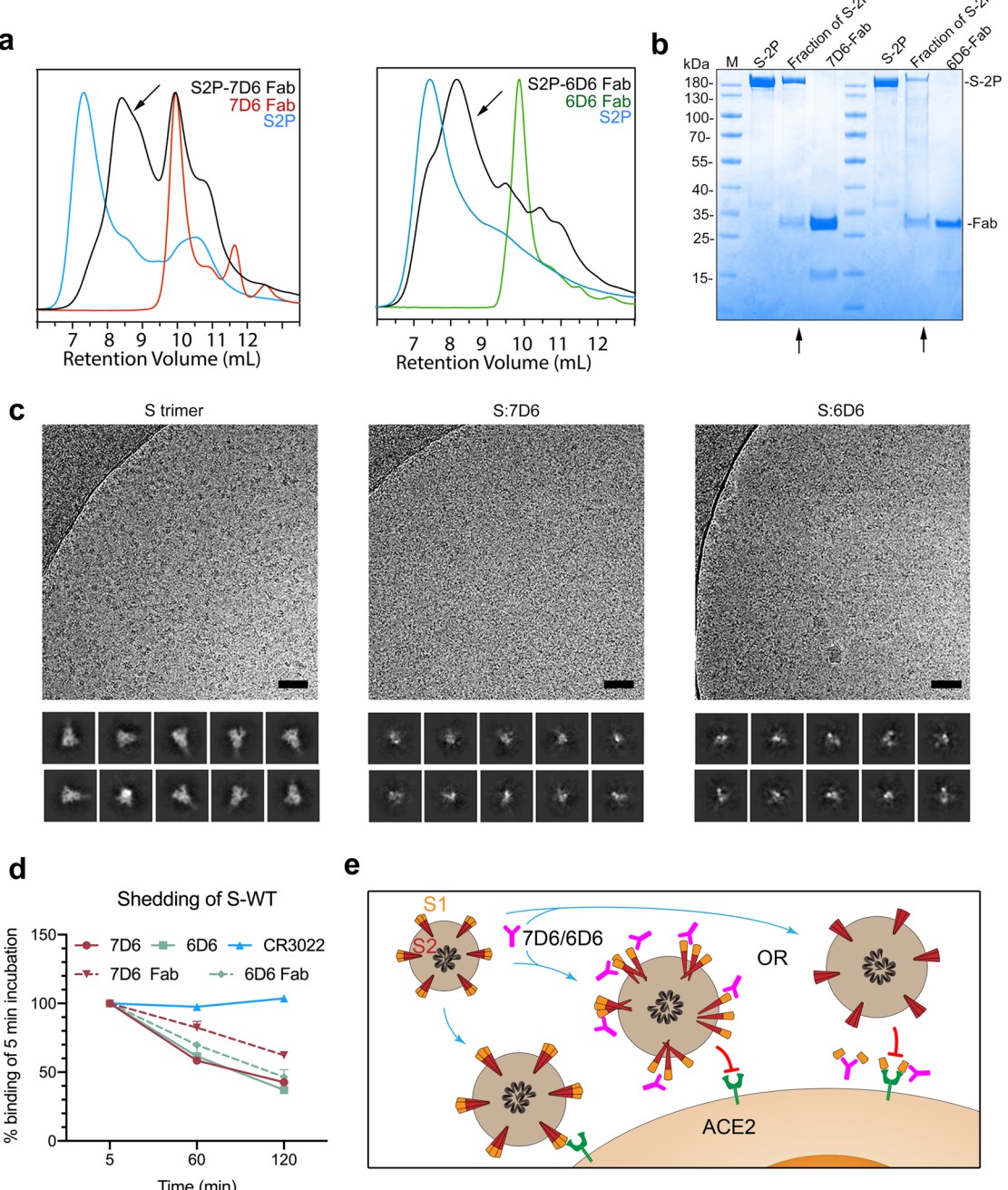

**Fig. 6 Neutralization mechanism revealed by biochemical and cryo-EM analyses. a** HPLC profiles of 7D6 and 6D6 binding to the S trimer. The black peak (arrow) indicates depolymerization of the spike. **b** SDS-PAGE analysis of the fractions harvested from the gel filtration chromatography of the mixture of S-2P and 7D6 or 6D6 Fab. The fractions are the peaks indicated by arrows in HPLC profile. **c** Cryo-EM micrographs (upper) and 2D analysis (lower) of S-2P and its immune complexes. Scale bar = 50 nm. **d** Shedding of S1 by IgG- and Fab forms of nAbs measured at 293 T cell-surface decorated with SARS-CoV-2 wild-type S protein by flow cytometry. **e** Two possible mechanisms of 7D6/6D6-mediated neutralization. First, 7D6/6D6 destabilizes SARS-CoV-2 spikes on the virus surface and the disordered spikes lose ability to engage ACE2 receptor. Or second, 7D6/6D6 binding triggers shedding off S1 moieties from S trimers, rendering viruses non-infectious, even though the free 7D6/6D6-bound S1 moieties could still engage ACE2. The experiments in (**b**) and (**c**) were performed twice with similar results. Source data are provided as a Source Data file.

Furthermore, the combined immunization strategy suggests that cross-neutralizing antibody might be readily produced by vaccination of convalescent COVID-19 patients who have recovered from infection by the original SARS-CoV-2 or some of its variants. The more potent cross-neutralizing antibodies might be isolated from those convalescent individuals and used to improve the efficacy and mutation resistance of therapeutic antibodies or cocktails against current and future variants.

A wide-spectrum vaccine is an appealing way to establish broad immunity and good tolerance against emerging variants. Identification of conserved sites and understanding antibody-mediated neutralization mechanisms are essential for design of

broadly protective vaccines[56,57,59]. Although 7D6 and 6D6 recognize conserved epitopes on the RBD distinct from those bound by most broadly protective antibodies and are relatively insensitive to the mutations (from the common, G614 parent strains) in the B.1.17, P.1, and B.1.351 variants, the 7D6 and 6D6 antibodies themselves would require further modifications—such as mouse-human chimeric grafting, humanized engineering and affinity maturation—to be candidates for clinical application. Their protective efficacy in vivo will also need to be verified. How best to expose the cryptic 7D6/6D6 site in immunogen design will likewise be required before contemplating any pre-clinical trials.

In summary, we have elicited SARS-CoV-2 cross-neutralizing antibodies in mice by a combined immunization strategy and shown that two cross-neutralizing antibodies, 7D6 and 6D6, bind tightly to the RBD and cross-neutralize SARS-CoV-2 and SARS-CoV by targeting a conserved cryptic site on the RBD. These two antibodies interfere with the neighboring NTD and appear to favor dissociation of S1 from the spike complex. The insensitivity of 7D6/6D6 to the mutations found in the RBDs of currently circulating SARS-CoV-2 variants suggests that 7D6 and 6D6 merit consideration as candidate therapeutic antibodies. The antigenic region described here is also a good target for rational design of pan-sarbecovirus vaccines.

## Methods

**Ethics statement**. All procedures in this study involving the authentic COVID-19 virus were performed in a biosafety level 3 (BSL-3) facility of the Shenzhen Third People's Hospital, China. The experimental protocols were approved by the Xiamen University Laboratory Animal Management Ethics Committee. All manipulations were strictly conducted in compliance with animal ethics guidelines and approved protocols.

**Cell lines**. Cell lines used in this study were obtained from the ATCC (H1299, BHK21, 293T, and Vero E6) or Thermo Fisher Scientific Inc. (CHO, sf9, and High Five cells). All cell lines used in this study were routinely tested for mycoplasma and found to be mycoplasma-free.

**Cloning, protein expression and purification of *Sarbecovirus* S-related proteins**. The SARS-CoV-2, SARS-CoV, MERS-CoV S-2P proteins, and/or RBDs were prepared as previously described[26]. In brief, the S genes (from the sequences of Genbank accession nos. NC_045512.2, NC_004718.3, and AFY13307.1 corresponding to SARS-CoV-2, SARS-CoV, and MERS-CoV, respectively) were synthesized and individually cloned into a baculovirus shuttle vector pAcgp67B (BD Biosciences, CA, USA) using Gibson assembly. The expression and purification of S-2P and RBD constructs were performed as described previously[26]. Hive Five cells (BTI-TN-5B1-4) (Thermo Fisher Scientific) were cultured in ESF921 medium (Expression Systems) and infected with recombinant virus at a multiplicity of infection (MOI) of 5 in the exponential growth phase ($2 \times 10^6$ cells/mL; 95% viability) at 28 °C for 72 h. The culture media was centrifuged at 7000$g$ for 20 min. The supernatant was dialyzed against phosphate-buffered saline (PBS), pH 7.4, purified with Ni-sepharose fast-flow 6 resin (GE Healthcare, Boston, USA), and eluted with 250 mM imidazole. The protein concentrations of the final purified samples were measured with Pierce BCA Protein Assay Kit (Thermo Fisher Scientific). The SARS-CoV-2 RBD mutants (N501Y, L452R, K417T/E484K/N501Y) were purchased from Sino Biological Inc. (Beijing, China).

**Monoclonal antibodies (mAbs)**. The immune scheme for mAb production was described by us elsewhere[26]. Mice were kept on a 12-h light/12-h dark cycle, at 22–24 °C and 30–70% humidity, with ad libitum access to food and water. MAbs were prepared following standard hybridoma technology, as previously described[60]. Fusion was performed 2 weeks after the final immunization. The resulting hybridomas were screened for the secretion of SARS-CoV-2- and SARS-CoV-specific mAbs using a S-2P binding assay. Hybridoma cells were cloned using limiting dilution at least three times, and positive clones were expanded and cultured in 75-cm[2] flasks. MAbs prepared by injecting hybridoma cells into the peritoneal cavities of pristine-primed BALB/c mice; ascites was collected after 9–12 days and stored at −20 °C. MAbs were purified from mouse ascites using protein A agarose columns (GE Healthcare).

**Expression and purification of IgG**. The variable domain genes of CR3022 (ref. [23]) and REGN10933 (ref. [32]) heavy and light chains were inserted into a pTT5 (Thermo Fisher Scientific) vector containing the constant region of the human IgG. The recombinant antibodies were expressed in Chinese hamster ovary (CHO) cells

through transient transfection and purified from culture media by affinity chromatography using MabSelect Sure resin (GE Healthcare).

**SDS-PAGE**. Protein samples were mixed with loading buffer (50 mM Tris pH 6.8, 2% SDS, 5% 2-mercaptoethanol, 0.01% bromophenol blue, 8% glycerol), boiled for 10 min, and subjected to sodium dodecyl sulfate-polyacrylamide gel electrophoresis (SDS-PAGE). Equal amounts of protein for each sample were loaded onto SDS-PAGE gels. The proteins were electrophoresed for 70 min at 120 V in a BioRad MINI-PROTEAN Tetra system (BioRad Laboratories, CA, USA), and the gel was stained with Coomassie Brilliant Blue R-250 (Bio-Rad) for 30 min at room temperature.

**Enzyme-linked immunosorbent assay**. Purified proteins were coated into the wells of 96-well microtiter plates at 100 ng/well in PBS and incubated at 37 °C for 4 h. The background was blocked with 1× enzyme dilution buffer (PBS + 0.25% casein + 1% gelatin + 0.05% proclin-300) at 37 °C for 2 h. Antibodies or sera at 2 μg/mL or 1:100, respectively, was threefold serially diluted, added to the wells (100 μL), and incubated at 37 °C for 1 h. A horseradish peroxidase (HRP)-labeled goat anti-mouse or goat anti-human antibody (Abcam) was used as the secondary antibody at 1:5000 for 30 min. Wells were washed again and the reaction catalyzed using $o$-phenylenediamine substrate at 37 °C for 10 min. The $OD_{450nm}$ (reference, $OD_{620nm}$) was measured on a microplate reader (TECAN, Männedorf, Switzerland) with a cut-off value of 0.1. The half-effective concentration ($EC_{50}$) or half-effective titers ($ET_{50}$) was calculated by sigmoid trend fitting using GraphPad Prism software (GraphPad Software, CA, USA).

**SARS-CoV-2 neutralization assay**. SARS-CoV-2 live virus focus reduction neutralization test (FRNT) was performed in a certified Biosafety level 3 laboratory, as previously described[5]. Neutralization assays against live SARS-CoV-2 were conducted using a clinical isolate (EPI_ISL_406594, WT, https://www.epicov.org/epi3/frontend#1903f; GWHBDSE01000000, B.1.351), previously obtained from a nasopharyngeal swab of an infected patient. Serial dilutions of tested antibodies were mixed with 50 μL of SARS-CoV-2 (100 focus forming units) in 96-well microwell plates and incubated at 37 °C for 1 h. Mixtures were then transferred to 96-well plates seeded with Vero E6 cells and allowed to absorb for 1 h at 37 °C. Inoculums were removed before adding the overlay media (100 μL MEM containing 1.6% carboxymethylcellulose). The plates were then incubated at 37 °C for 24 h. Overlays were removed and then cells were fixed with 4% paraformaldehyde solution for 30 min, and permeabilized with Perm/Wash buffer (BD Biosciences) containing 0.1% Triton X-100 for 10 min. Cells were incubated with rabbit anti-SARS-CoV-2 NP IgG (Sino Biological, Inc.) for 1 h at room temperature followed by HRP-conjugated goat anti-rabbit IgG (H + L) antibody (TransGen Biotech, Beijing). The reactions were developed with KPL TrueBlue Peroxidase substrates (Seracare Life Sciences Inc.). The numbers of SARS-CoV-2 foci were calculated using an EliSpot reader (Cellular Technology Ltd).

**Pseudotype LV-based neutralization test**. Antibodies were tested against lentiviral pseudotyping particles (LVpp) bearing the SARS-CoV-2 spike antigen based on H1299-ACE2hR cells, as described previously[27]. In briefly, SARS-CoV-2 LVpp were generated by co-transfection of a lentiviral packaging plasmid (psPAX2, Addgene), a SARS-CoV-2 spike expression plasmid (containing codon-optimized spike gene derived from the strain of MN908947.3 (WT) or EPI_ISL_601443 (B.1.1.7, https://www.epicov.org/epi3/frontend#2c197c) or EPI_ISL_700428 (B.1.351, https://www.epicov.org/epi3/frontend#31cdb6) or EPI_ISL_792680 (P.1, https://www.epicov.org/epi3/frontend#16776d) and a green fluorescent protein (mNeonGreen) reporter vector (pLvEF1α-mNG, carrying EF1α promoter-driven mNeonGreen expressing cassette) in 293 T cells. Infection and neutralization assays were performed on H1299-ACE2hR cells, which stably over-expressed human ACE2 (enabling it is highly susceptible to SARS-CoV-2 virus) and nuclear-localized RFP (H2B-mRuby3, allowing accurate cell counting) based on H1299 cells. For ppNAT tests, serially diluted antibodies were incubated with LVpp inoculum (0.5 TU/cell) for 1 h. Subsequently, the mixtures were incubated with the cells, which had been pre-seeded in 96-well cell culture plates with an optically clear bottom. After 36-h incubation, the plates were imaged by using Opera Phenix or Operetta CLS high-content equipment (PerkinElmer). For quantitative determination, fluorescence images were analyzed by Columbus Software 2.5.0 (PerkinElmer), the numbers of mNeonGreen (+) cells per well were calculated to indicate the infection performance, and the total cell numbers per well were also counted to normalize the readouts. The reduction (%) on mNeonGreen (+) cells of the plasma-treated well in comparison with control-well was calculated to show the neutralization activity. The ppNAT titer of each sample were expressed as the maximum dilution concentration required to achieve infection inhibition by 50% ($IC_{50}$). The $IC_{50}$ value was determined by the 4-parameter logistic (4PL) regression using GraphPad Prism v8.0.

**Pseudotype VSV-based neutralization test**. The VSV-based neutralization test was carried out as described[28]. The cultured supernatant of the monoclonal hybridoma cells and gradient-diluted purified antibodies were mixed with diluted VSV-SARS-CoV-2-Sdel18 virus (MOI = 0.05) and incubated at 37 °C for 1 h. All

samples and viruses were diluted with 10% FBS-DMEM. The mixture was added to pre-seeded BHK21-hACE2 cells. After incubating for 12 h, fluorescence images were obtained with Opera Phenix or Operetta CLS equipment (PerkinElmer). For quantitative analysis, fluorescence images were analyzed using the Columbus system (PerkinElmer), and the numbers of GFP-positive cells for each well were counted to represent infection performance. The reduction (%) in the number of GFP-positive cells in mAb-treated wells compared with that in nontreated control wells were calculated to show the neutralizing potency.

**Size-exclusive chromatography**. All high-purity RBD, Fab, and immune complex proteins were subjected to HPLC (Waters; Milford, MA) analysis using a TSK Gel G5000PWXL7.8 × 300-mm column (TOSOH, Tokyo, Japan) equilibrated in PBS, pH 7.4. The system flow rate was maintained at 0.5 mL/min and eluted proteins were detected at 280 nm.

$K_D$ **determination**. $K_D$ values were determined by SPR technology using a Biacore 8K instrument (GE Healthcare). The S-2P or RBD was amine-coupled to a CM-5 sensor chip for use. Antibodies was then captured on the sensor surface at a flow rate of 30 μL/min in PBS-P + buffer (0.2 M phosphate buffer with 27 mM KCl, 1.37 M NaCl, and 0.5% Surfactant P20 (Tween 20)). The antibodies were tested using serially diluted concentrations (200, 150, 100, 75, 50, 37.5, 25, 18.75, 12.5, and 9.375 nM). The flow durations were 200 s for the association stage and 10 min for dissociation. Association rates ($k_a$), dissociation rates ($k_d$), and affinity constants ($K_D$) were calculated using BIAcore evaluation software.

**Competition assay**. A competition assay was carried out to investigate the binding mode of the monoclonal antibodies with hACE2 (or between two antibodies) using SPR technology in a Biacore 8K instrument (GE healthcare). All experiments were performed at 25 °C, and the biosensors were pre-equilibrated in PBS-P + buffer (0.2 M phosphate buffer with 27 mM KCl, 1.37 M NaCl, and 0.5% Surfactant P20 [Tween 20]) for 10 min. Antibodies (first protein) at 3000 nM were loaded onto the biosensors for 500 s, followed by flow of the second interacting protein (hACE2 or the second antibody), also at 3000 nM for 500 s. The unblocked pattern of the RBD with buffer was used as a control.

**Preparation, crystallization, and structure determination of immune complexes**. 7D6, 6D6, and 16D8 Fabs were prepared by papain digestion of the mAb and purified with Protein A (GE Healthcare). The SARS-CoV-2 RBD was mixed with each Fab in a 1:1.2 molar ratio and incubated at 37 °C for 2 h. The immune complex was further purified to remove any excess Fab by gel filtration on a Superdex 200 Increase column (GE Healthcare) in 10 mM Tris pH 8.0 with 50 mM NaCl. The complex was concentrated to ~7.5 mg/mL for crystallization.

The crystallization was performed using sitting-drop vapor diffusion in the screening stage and hanging drop in microseeding optimization at 20 °C. Crystals of the 7D6:RBD complex were grown in 0.2 M potassium dihydrogen phosphate with 20% PEG3350, whereas crystals of the 6D6:RBD complex were grown in 0.2 M sodium thiocyanate with 20% PEG3350. Crystal growth took about 7 days before final data collection. Crystals were cryo-protected in reservoir solution supplemented with 30% glycerol at 100 K before collection of the diffraction data. Diffraction data were collected at Shanghai Synchrotron Radiation Facility (SSRF) beamline BL17U1 using a DECTRIS EIGER X 16M Detector (wavelength, 0.97919 Å). The diffraction data were auto-processed with XDS software by aquarium pipeline[61]. The complex structures were determined by the molecular replacement method using Phaser[62]. The search model for RBD, 7D6 Fab, and 6D6 Fab phasing are COVID-19 RBD structure (PDB no. 6M0J: https://www.rcsb.org/structure/6M0J), Fab structure in PDB no. 6RCO, and Fab structure in PDB no. 1WEJ, respectively. The resulting models were manually built in COOT[63], refined with PHENIX[64] and analyzed with MolProbity[65]. In brief, one round of rigid-body refinement was performed after molecular replacement phasing. The refined models were manually modified in COOT; coordinates and individual B factors were refined in reciprocal space. TLS refinement was performed in the later stages with auto-searched TLS groups in PHENIX, which were listed in REMARK 3 sections in the deposited cif files. Data collection and structure refinement statistics are summarized in Table S3. All figures were prepared with PyMoL Molecular Graphics System (https://pymol.org).

**Sequence and conservation analysis of epitopes**. SARS-CoV-2 S gene mutations were calculated based on those within GISAID (https://www.gisaid.org) on 11 July 2021 ($n = 2,216,094$). SARS-CoV S genes were sourced from ViPR (https://www.viprbrc.org) and NCBI (https://www.ncbi.nlm.nih.gov) using only those deposited before December 2019 to exclude SARS-CoV-2 (search criteria: SARS-related coronavirus, full-length genomes, all host; $n = 77$, performed on 11 July 2021). Pangolin sequences were sourced from GISAID ($n = 6$). Reported for percent conservation are the number of sequences with an identical change at a position divided by the total number of sequences (include *Sarbecovirus* clade 1, 2, 3 (ref. [29])). The conservation figure indicating the sequence conservation was generated using PyMoL Molecular Graphics System.

**Preparation and data collection of Cryo-EM samples**. Aliquots (3 μL) of 0.5 mg/mL purified S-2P protein or its immune complexes were loaded onto glow-discharged (60 s at 20 mA) holey carbon Quantifoil grids (R1.2/1.3, 200 mesh, Quantifoil Micro Tools) using a Vitrobot Mark IV (Thermo Fisher Scientific) at 100% humidity and 4 °C. Data were acquired using the EPU software on an FEI Tecnai F30 transmission electron microscope (Thermo Fisher Scientific) operated at 300 kV and equipped with a Thermo Fisher Falcon-3 direct detector. Images were recorded in the 39-frame movie mode at a nominal magnification of 93,000× with a pixel size of 1.12 Å on the sample level and an underfocus range of 1.5–2.8 μm. The total electron dose was set to $30e^- $ Å$^{-2}$ and the exposure time was 1.0 s.

**Cryo-EM data processing**. Movie frame alignment and contrast transfer function estimation of each aligned micrograph were carried out with the programs of Motioncor[66] and Gctf[67]. Particles were picked by the "Template picker" session of CryoSPARC v2 (ref. [68]). Two rounds of reference-free 2D classification were performed, and well-defined particle images were selected.

**Spike shedding assay**. The spike shedding assay was performed as described by Yan et al.[69]. In brief, plasmids encoding SARS-CoV-2 wild-type S were transfected into HEK293T cells. Cells samples were prepared in multiples for serial incubation with IgG or Fabs (1 mg/mL) at 37 °C for 120, 60, and 5 min. Immediately after the incubation time, cells were transferred to ice then thoroughly washed with ice-cold PBS and 2% FBS. Samples were then stained with anti-mouse IgG (H + L) Alexa Flour 647 (Thermo Fisher) for 30 min. After thorough washes with ice-cold PBS and 2% FBS, samples were resuspended and analyzed by FACS Calibur (BD Biosciences, USA) and FlowJo 10 software (FlowJo, USA). Binding at each time point was determined by the percentage of positive cells in the selected gates and normalized correlated to that at the time point of 5 min (Fig. S5).

**Reporting summary**. Further information on research design is available in the Nature Research Reporting Summary linked to this article.

## Data availability
The coordinates and structure factors for 7D6:RBD, 6D6:RBD have been deposited in the Protein Data Bank (accession nos. 7EAM and 7EAN). The amine acid sequence of 7D6 and 6D6 in this study are provided in Table S6 in Supplementary Information. Source data are provided with this paper.

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

## Acknowledgements

This work was supported by grants from the National Natural Science Foundation of China (grant nos. 82001756, 81991491 and U1705283), the National Key Research and Development Program of China (grant no. 2020YFC0842600), the Science and Technology Major Projects of Xiamen (grant no. 3502Z20203023). We thank Dr. Huan Zhou (beamline BL17U1 at the Shanghai Synchrotron Radiation Facility) for the assistance in X-ray data collection and processing. We would like to thank Guangdong Center for Human Pathogen Culture Collection (GDPCC) of China and Guangdong Provincial Center for Disease Control and Prevention of China for providing the authentic SARS-CoV-2 (B.351) for neutralization assay.

## Author contributions

Z.H.Z., Z.Z., Y.G., S.L. and N.X. designed the study. T.L., W.X., Q. Zheng, S.S., C.Y., H.X., S.Z., M.H., Yali Zhang, Yuyun Zhang., H.S., Y.H., T.D. X.C., J.L., S.W., L.Z. and T. Chen performed experiments. T.L., W.H., Q. Zheng, H.Y., Y.G., S.L., and N.X. analyzed data. T.L., Q. Zheng, J.S.M., Z.H.Z., Y.G. and S.L. wrote the manuscript. T.L., W.X., Q. Zheng, Y.W., T. Cheng, T.Z., Q.Y., Q. Zhao, Z.H.Z., J.Z., Y.G., S.L. and N.X. participated in discussion and interpretation of the results. All authors contributed to experimental design.

## Competing interests

The authors declare no competing interests.
