## [Peer Review File · Nature Communications]

Cross-neutralizing antibodies bind a SARS-CoV-2 cryptic site and resist circulating variantsReviewers' Comments:

Reviewer #1:

Remarks to the Author:

Review of Li et al.

This MS reports an effort to obtain broad-spectrum coronavirus neutralizing antibodies by immunizing mice with SARS-CoV-2 spike combined with MERS-CoV RBD as well as SARS-CoV-2 spike alone. They then selected hybridomas for cross reactivity with SARS-CoV-1 and -2 spikes (I realize that the standard notation is SARS-CoV, but I find it confusing when seen together with SARS-CoV-2, so I'll use the HIV convention), obtaining 5 and 10 lead Abs from the single and combined immunizations respectively. Eight of the latter bound the RBD, of which 6 cross-neutralized (one of the former bound RBD and cross neutralized also). Selecting three candidates for detailed characterization -- 7D6, 6D6 and 16D8 -- they showed that the first two competed with each other for RBD binding and determined their crystal structures bound with SARS-CoV-2 RBD. The combined epitopic region covers residues 347-357 and 457-471 and hence requires the "up" RBD configuration for binding, as the corresponding surface is occluded in the "down" configuration. None of the mutations in recently characterized variants affect that region. Finally, the authors show that the Fabs appear to stimulate dissociation of the spike, presumably into S1 and S2 (assuming that their S2P preparation is cleaved, or mostly cleaved, into the two principal fragments).

The MS contains interesting and worthwhile information, even if the immunodominant epitopic regions for mice could easily be somewhat different from those for humans and hence detailed comparisons with the epitopic distribution for human antibodies are not in order, nor is the classification in the paragraph starting at line 176. Should the authors propose to revise the MS for resubmission to this journal, the reviewer asks for responses to the following points.

Specific points.

1. As the immunization that yielded most of the cross-binders included MERS-CoV RBD (rather than spike), the authors cannot conclude (line 102) that the RBD is relatively more immunogenic for cross-reactive responses than is S2. The sentence beginning on line 101 should be deleted.
2. The attempted classification (lines 176ff) is inappropriate. Not only does it compare apples and oranges (murine vs. human responses), but it is too finely divided because it picks just a few RBD-directed Abs for comparison. Taking, for example, all the published structures (at least 50 at last count -- perhaps more) of human Fab complexes (with either spike or RBD) that target the ACE2 binding surface (the RBM) and superposing them all on the RBD, one gets a continuous fan of structures that includes both Regeneron Abs. Trying to subclassify just isn't helpful. Even if Fabs at the extremes of the fan do not overlap, there is a continuum of intermediates. The structures of 6D6 and 7D6 as shown in Fig. 3 might correspond to the recently described "RBD-1" category of Tong et al (bioRxiv 2021.03.10.434840). Although that category includes S309, it appears to extend around as far as the apparent 7D6 and 6D6 epitopes. This reviewer believes that the categorization in the Tong et al MS is about as finely divided as one should go, and even those authors point out that the RBD Abs in the three categories they identify shade into each other when tested by cross competition for binding.
3. The authors state (line 189) that S309 "neutralizes" (presumably they mean protects) by, inter alia, ADCC and ADCP, but Pinto et al (their reference) actually claim potent infectivity neutralization in cell culture (absent effector mechanisms) and only indirectly infer that ADCC or ADCP might contribute to protection in vivo.
4. The absence of an SDS-PAGE gel (or even a Western with distinct S1 and S2 specific Abs) in Fig. 6 greatly weakens any conclusion. The most likely explanation for the micrographs is "shedding", i.e.,

dissociation of S1 from S2, and conversion of the latter to the postfusion conformation (as concluded for S309, although not very rigorously either) in their ref 24. Indeed, the authors come to this conclusion in the first paragraph of the discussion. As they also point out, even if the cleavage has not occurred, or if it is incomplete, S1 can move away from S2 if suitably destabilized, leading to some sharp features and some fuzzy and unclassifiable ones. Thus, first, we need to know whether the S2P was cleaved. Second, as Choe, Farzan and co-workers have shown, the G614 variant sheds much less readily than does the original Wuhan isolate. From which was the S2P used here derived? If the latter, and if the cleavage was at least 30-50% complete, the effect of the Fab would have been to accelerate shedding (i.e., dissociation), consistent with the apparent relative instability of the "3-up" configuration. Antibodies that destabilize the spike might well be useful, but the analysis should be carried out more thoroughly and more rigorously, to reach a conclusion that might allow one of these Abs to be used (in further work, not required here) to ask whether Abs that induce shedding can actually protect from otherwise measurable infection in a suitable animal model. Absent that more complete analysis, any conclusions from an animal model would be suspect.

5. Extrapolating to the conclusions in the paragraph starting on line 229 is going too far. The results of a single, limited, combined immunization in mice should not be a platform for wild speculation about pan-Sarbecovirus vaccine regimens. Any analysis of potential pan-virus immunity would have to consider B cell memory and the potential for affinity maturation either toward or away from cross-neutralization.

Reviewer #2:

Remarks to the Author:

The manuscript reports the structure of two antibodies recognizing a previously undescribed site on the SARS2 RBD.

I found it overall very interesting and with significant implications for SARS2 therapy.

Impact would be increased by in-animal testing; by a more thorough investigation of binding to the full S trimer and of the mechanism of neutralization. These would be a plus rather than a necessity for publication.

Specific comments, all considered minor except the two indicated by a *

- Looking at figure 5C and 5K, the epitope seems only partially obstructed by the NTD. I am not that convinced to call that a 'cryptic site' and think that 'partially inaccessible' would be a more correct description.

However, I am ready to accept the Authors' term if they feel strongly about it.

*- If 7D6 and 6D6 bind with similar orientation (line 139), how do the Authors explain the different behaviour when comparing binding to RBD vs S-trimer? 7D6 binding is stronger to RBD than S but it does not seem to be the case for 6D6 according to the text.

*- Epitope conservation and possibly resistance to mutations is a critical feature for antibody-based immunotherapy.

The sequence alignment shown in S2 should be moved to the main paper (as a panel in Fig 3 I suggest) but, more importantly, it should be expanded to other sequences of both SARS2 and other CoVs. The (recent) variants of concern should be better highlighted, ideally including current nomenclature (alpha, delta etc).

The claim of 'highly conserved across sarbecovirus isolates' (line 150) is not clearly supported by the limited sequence analysis shown.

- I find Figure 4B a bit unattractive; the 3rd panel in particular.
I suggest changing visualization style, possibly to a table and standard neutralization curves as shown in S4.

- Line 84

Following immunization of what?

I recommend adding 'animal' or whatever is appropriate, even if mice are indicated later in the manuscript

- All the SPR panels in Figure 2E can go in supplementary without impacting the manuscript. However, I recommend focusing on the important points and showing (in a table or else) that i) 7D6 and 16D8 bound RBD with higher affinity than S trimer and ii) 6D6 has higher affinity for SARS2 than SARS

- When performing cross-competition by SPR (Fig 2G) I recommend using much shorter dissociation time before the second injection.

Units of measurement must be shown on the x axis in Fig 2 (all SPR panels lack them).

- Line 135

It is strange to read 'we determined chemical bonds' when describing the x-ray structure of an Ab/Ag complex. I suggest rewording.

Response to Reviewer Comments on the manuscript [NCOMMS-21-21294]:

We thank the two reviewers for recognizing the merit of our work and for their suggestions to improve our manuscript. We have fully addressed the comments with appropriate additional experiments and analyses. To facilitate the navigation of this document, we have copied the reviewers' comments verbatim in **blue** and typed our responses in **black**.

Reviewer #1 (Remarks to the Author):

This MS reports an effort to obtain broad-spectrum coronavirus neutralizing antibodies by immunizing mice with SARS-CoV-2 spike combined with MERS-CoV RBD as well as SARS-CoV-2 spike alone. They then selected hybridomas for cross reactivity with SARS-CoV-1 and -2 spikes (I realize that the standard notation is SARS-CoV, but I find it confusing when seen together with SARS-CoV-2, so I'll use the HIV convention), obtaining 5 and 10 lead Abs from the single and combined immunizations respectively. Eight of the latter bound the RBD, of which 6 cross-neutralized (one of the former bound RBD and cross neutralized also). Selecting three candidates for detailed characterization -- 7D6, 6D6 and 16D8 -- they showed that the first two competed with each other for RBD binding and determined their crystal structures bound with SARS-CoV-2 RBD. The combined epitopic region covers residues 347-357 and 457-471 and hence requires the "up" RBD configuration for binding, as the corresponding surface is occluded in the "down" configuration. None of the mutations in recently characterized variants affect that region. Finally, the authors show that the Fabs appear to stimulate dissociation of the spike, presumably into S1 and S2 (assuming that their S2P preparation is cleaved, or mostly cleaved, into the two principal fragments).

The MS contains interesting and worthwhile information, even if the immunodominant epitopic regions for mice could easily be somewhat different from those for humans and hence detailed comparisons with the epitopic distribution for human antibodies are not in order, nor is the classification in the paragraph starting at line 176. Should the authors propose to revise the MS for resubmission to this journal, the reviewer asks for responses to the following points.

Response: As detailed below, we have addressed the issues raised. The standard notation "SARS-CoV" is now used throughout the revised manuscript.

Specific points:

Comment 1: As the immunization that yielded most of the cross-binders included MERS-CoV RBD (rather than spike), the authors cannot conclude (line 102) that the RBD is relatively more immunogenic for cross-reactive responses than is S2. The sentence beginning on line 101 should be deleted.

Response: We agree and have deleted this sentence. (Page 5, line 101).

Comment 2: The attempted classification (lines 176ff) is inappropriate. Not only does it compare apples and oranges (murine vs. human responses), but it is too finely divided because it picks just a few RBD-directed Abs for comparison. Taking, for example, all the published structures (at least 50 at last count -- perhaps more) of human Fab complexes (with either spike or RBD) that target the ACE2 binding surface (the RBM) and superposing them all on the RBD, one gets a continuous fan of structures that includes both Regeneron Abs. Trying to subclassify just isn't helpful. Even if Fabs at the extremes of the fan do not overlap, there is a continuum of intermediates. The structures of 6D6 and 7D6 as shown in Fig. 3 might correspond to the recently described "RBD-1" category of Tong et al (bioRxiv 2021.03.10.434840). Although that category includes S309, it appears to extend

around as far as the apparent 7D6 and 6D6 epitopes. This reviewer believes that the categorization in the Tong et al MS is about as finely divided as one should go, and even those authors point out that the RBD Abs in the three categories they identify shade into each other when tested by cross competition for binding.

Response: As suggested, the 7D6 and 6D6 are now grouped into the “RBD-1” category together with S309 (class 1 in our manuscript). To narrow down the vast of SARS-CoV-2 antibody information, we only included in our study those antibodies that have structural epitope information for classification. We have revised the description in the main text (Page 8, line 174) and updated the Fig. 4.

Comment 3: The authors state (line 189) that S309 “neutralizes” (presumably they mean protects) by, *inter alia*, ADCC and ADCP, but Pinto et al (their reference) actually claim potent infectivity neutralization in cell culture (absent effector mechanisms) and only indirectly infer that ADCC or ADCP might contribute to protection *in vivo*.

Response: We have deleted this sentence in the revised manuscript. (Page 8, line 176)

Comment 4: The absence of an SDS-PAGE gel (or even a Western with distinct S1 and S2 specific Abs) in Fig. 6 greatly weakens any conclusion. The most likely explanation for the micrographs is “shedding”, i.e., dissociation of S1 from S2, and conversion of the latter to the postfusion conformation (as concluded for S309, although not very rigorously either) in their ref 24. Indeed, the authors come to this conclusion in the first paragraph of the discussion. As they also point out, even if the cleavage has not occurred, or if it is incomplete, S1 can move away from S2 if suitably destabilized, leading to some sharp features and some fuzzy and unclassifiable ones. Thus, first, we need to know whether the S2P was cleaved. Second, as Choe, Farzan and co-workers have shown, the G614 variant sheds much less readily than does the original Wuhan isolate. From which was the S2P used here derived? If the latter, and if the cleavage was at least 30-50% complete, the effect of the Fab would have been to accelerate shedding (i.e., dissociation), consistent with the apparent relative instability of the “3-up” configuration. Antibodies that destabilize the spike might well be useful, but the analysis should be carried out more thoroughly and more rigorously, to reach a conclusion that might allow one of these Abs to be used (in further work, not required here) to ask whether Abs that induce shedding can actually protect from otherwise measurable infection in a suitable animal model. Absent that more complete analysis, any conclusions from an animal model would be suspect.

Response: We appreciate the reviewer’s constructive criticisms about how to elucidate neutralization mechanism. We now include additional data to address the issues raised.

Firstly, the SDS-PAGE in the former Fig. S5 has now been moved to the updated Figure 6, which confirms the cleavage did not happen at the S2P proteins while binding Fab and undergoing dissociation. This biochemical result is reasonable, as the S2P was designed with two proline substitutions at residues 986 and 987 for trimer-stabilization and the mutation of “AGAG” at the furin cleavage site (residues 682-685) for cleavage silence. The S2P trimer was produced in insect cells and well-characterized in our previous study (Li et al. EMI. 2020.). Secondly, we measured the shedding of wild-type SARS-CoV-2 spike upon binding of 7D6, 6D6 or control CR3022 Fab as the assay described by Yan et al (2021, Cell Research), where the full-length S was expressed and transferred at the surface of 293FT cells and allowed to be quantified by flow cytometry. In the new Fig. 6D, both full-length and Fab forms of antibodies 7D6 and 6D6 do trigger the spike shedding up to 63% after incubating with cells for 120 min, which is more potently than CR3022. Although our

results confirm the antibody-induced spike shedding in vitro, we concur with the reviewer that an appropriate animal model should be applied to further establish the mechanism in vivo, which will be pursued in our future work.

Comment 5: Extrapolating to the conclusions in the paragraph starting on line 229 is going too far. The results of a single, limited, combined immunization in mice should not be a platform for wild speculation about pan-Sarbecovirus vaccine regimens. Any analysis of potential pan-virus immunity would have to consider B cell memory and the potential for affinity maturation either toward or away from cross-neutralization.

Response: We agree and now have toned down our immunization regimen for cross-neutralization immunity and emphasize B cell memory. It now reads: “Vaccination regimens through combined and/or sequential immunization strategy might provide cross-immunity upon SARS-CoV-2 and SARS-CoV as well as circulating variants by virtue of the cross-neutralizing antibody response, for instance from the 7D6/6D6 site that is highly conserved across Sarbecovirus isolates. Moreover, the potential for affinity maturation in the development of human B cell memory during long-term immunization should be considered toward the aim for cross-immunity.” (Page 11, line 234).

Reviewer #2 (Remarks to the Author):

The manuscript reports the structure of two antibodies recognizing a previously undescribed site on the SARS2 RBD. I found it overall very interesting and with significant implications for SARS2 therapy. Impact would be increased by in-animal testing; by a more thorough investigation of binding to the full S trimer and of the mechanism of neutralization. These would be a plus rather than a necessity for publication.

Response: We thank the reviewer for recognizing the merit of our work. We have now added new data on spike shedding from full S trimer-expressed 239FT cells, which further supports our proposed mechanism for the neutralization (see details in our answer to Comment 4 below). We agree that in-animal testing is very desirable and will follow up in our future work.

Specific comments, all considered minor except the two indicated by a *

Comment 1: Looking at figure 5C and 5K, the epitope seems only partially obstructed by the NTD. I am not that convinced to call that a 'cryptic site' and think that 'partially inaccessible' would be a more correct description. However, I am ready to accept the Authors' term if they feel strongly about it.

Response: Yes, the epitope of 7D6 and 6D6 is partially obstructed by the adjacent NTD, similar to CR3022 that is partially concealed by the neighboring RBD, see **Fig. R1** left (Yuan et al. 2020. Science), where the authors called the epitope as “cryptic site” (Yuan et al. 2020. Science). While the suggested

term “partially inaccessible” is correct, we wish to keep the “cryptic site” term as used in the literature.

Comment 2*: If 7D6 and 6D6 bind with similar orientation (line 139), how do the Authors explain the different behaviour when comparing binding to RBD vs S-trimer? 7D6 binding is stronger to RBD than S but it does not seem to be the case for 6D6 according to the text.

Response: We now have added the following sentence to spell out a possible reason for the discrepancy of affinity to RBD/S2P between these two antibodies, “Despite of the similar binding orientation, 7D6 shows high affinity to RBD than to S2P, whereas 6D6 confers comparable affinities (Fig. 2E), which might be due to the epitope of 7D6 is more inaccessible in the structural context of S trimer than that of 6D6 (Fig. 5C, 5G, 5K and 5O).” (Page 10, line 222)

Comment 3*: Epitope conservation and possibly resistance to mutations is a critical feature for antibody-based immunotherapy. The sequence alignment shown in S2 should be moved to the main paper (as a panel in Fig 3 I suggest) but, more importantly, it should be expanded to other sequences of both SARS2 and other CoVs. The (recent) variants of concern should be better highlighted, ideally including current nomenclature (alpha, delta etc). The claim of 'highly conserved across sarbecovirus isolates' (line 150) is not clearly supported by the limited sequence analysis shown.

Response: Following the suggestion, we have moved the sequence alignment figure (the former Fig. S2) to the updated Fig. 3I with the addition of recent variants (names including alpha, delta nomenclature). The phrase “highly conserved across sarbecovirus isolates” is based on the result in Table S4, where the sequences of SARS-CoV-2 isolates (n=2,216,094), SARS-CoV and other SARS-related CoVs (N=83) include clade 1, 2, 3 (deposited in GISAID, ViPR, NCBI database, see methods) were used for alignment to identify highly conserved sequences.

Comment 4: I find Figure 4B a bit unattractive; the 3rd panel in particular. I suggest changing visualization style, possibly to a table and standard neutralization curves as shown in S4.

Response: As suggested, the original Figure 4B is now reformatted as a table, please refer to the updated Figure 4B.

Comment 5: Line 84. Following immunization of what? I recommend adding 'animal' or whatever is appropriate, even if mice are indicated later in the manuscript

Response: Revised as suggested. (Page 4, line 84)

Comment 6: All the SPR panels in Figure 2E can go in supplementary without impacting the manuscript. However, I recommend focusing on the important points and showing (in a table or else) that i) 7D6 and 16D8 bound RBD with higher affinity than S trimer and ii) 6D6 has higher affinity for SARS2 than SARS

Response: As suggested, we moved the SPR panels to supplementary materials and listed the kinetic constants in a table (see the updated Figure 2E).

Comment 7: When performing cross-competition by SPR (Fig 2G) I recommend using much shorter dissociation time before the second injection. Units of measurement must be shown on the x axis in Fig 2 (all SPR panels lack them).

Response: The units of measurement are now indicated on the x axis in the updated Fig. 2. The timepoint of ~200 sec for the second injection in cross-competition SPR assay was set according to the dissociation curve in the affinity data (see new Fig. S2). Overall, the three antibodies have relatively slow dissociation rate of about 1×10^{-5} /s; it may take longer time to reach a stable association/disassociation equilibrium than other antibodies with fast dissociation. Nonetheless, in

future studies, we will consider the duration at the first dissociation to see if there is any impact of dissociation time on competition assay.

Comment 8: Line 135. It is strange to read 'we determined chemical bonds' when describing the x-ray structure of an Ab/Ag complex. I suggest rewording.

Response: We changed “determined chemical bonds” to “characterized the details of binding” (Page 6, line 133).

Reviewers' Comments:

Reviewer #1:

Remarks to the Author:

Review of Li, Li et al.

The authors have revised the MS paying attention to most of my concerns. I attach a marked-up copy of the MS with extensive copy editing (main-text only -- the authors can see what I've done and make any necessary changes in the figure captions and methods), for three reasons. (1) There were many corrections needed to conform to standard, scientific English -- too many just to list. (2) There were several sentences that were simply incomprehensible: I hope that I have understood them well enough to provide an understandable, standard-English version. (3) There were some phrases or sentences that needed to be deleted because they were either inappropriate or wrong. These last I have deleted or revised and provided a comment to explain why. I left in place one bit of wry humor ("well-publicized case" and the NYT reference), which I think all readers of a paper like this one will enjoy (I certainly did -- thanks!).

The one remaining substantive point concerns the affinities in Figure 2, which I failed to examine closely enough on the first review (apologies). I'm concerned that the K_d for RBD binding (which is clearly very low -- i.e., high affinity -- so there's no argument about the qualitative description) is artifactually tiny (single-digit picomolar) because of an essentially immeasurably slow off rate (assuming no SPR mass transport artifact). Vicinal rebinding of the divalent IgG may be one contribution. As the experiments appear to have been done by standard methods, with appropriate IgG dilutions, I simply urge the authors to check carefully. Because they make no particular quantitative claims, and because similar (less carefully described) experiments are all over the literature, I do not believe that they should be asked to repeat any of the experiments.

The footprints of 7D6 and 6D6, from Fig. 3, appear to overlap the epitope of G32R7, an RBD-1 ("class 1" in the current authors' nomenclature) antibody and to border the epitope of S309 (the IgGs would probably compete). Thus, the authors' assignment of these two antibodies to this category is correct.

Reviewer #2:

None

Response to Reviewer Comments on the manuscript [NCOMMS-21-21294A]:

We thank the reviewer #1 for second round of review. We appreciate his/her agreement with our revision and extensive copy editing throughout the main text, which substantially improves our manuscript. Below shows our response (in **black**) to the reviewers' comments verbatim (in **blue**).

Reviewer #1 (Remarks to the Author):

The authors have revised the MS paying attention to most of my concerns. I attach a marked-up copy of the MS with extensive copy editing (main-text only -- the authors can see what I've done and make any necessary changes in the figure captions and methods), for three reasons. (1) There were many corrections needed to conform to standard, scientific English -- too many just to list. (2) There were several sentences that were simply incomprehensible: I hope that I have understood them well enough to provide an understandable, standard-English version. (3) There were some phrases or sentences that needed to be deleted because they were either inappropriate or wrong. These last I have deleted or revised and provided a comment to explain why. I left in place one bit of wry humor ("well-publicized case" and the NYT reference), which I think all readers of a paper like this one will enjoy (I certainly did -- thanks!).

Response: We indeed appreciate the reviewer's comment and cope editing to our manuscript. We have incorporated all his/her revision in the final version.

The one remaining substantive point concerns the affinities in Figure 2, which I failed to examine closely enough on the first review (apologies). I'm concerned that the K_d for RBD binding (which is clearly very low -- i.e., high affinity -- so there's no argument about the qualitative description) is artifactually tiny (single-digit picomolar) because of an essentially immeasurably slow off rate (assuming no SPR mass transport artifact). Vicinal rebinding of the divalent IgG may be one contribution. As the experiments appear to have been done by standard methods, with appropriate IgG dilutions, I simply urge the authors to check carefully. Because they make no particular quantitative claims, and because similar (less carefully described) experiments are all over the literature, I do not believe that they should be asked to repeat any of the experiments.

Response: We agree with the reviewer's concern regarding to the very low off-rate in SPR. Indeed, we generated the data by standard method with minimized mass transport effect. The high affinity is reasonable given bivalent binding or vicinal rebinding of IgG to RBDs i.e. avidity, which could be checked by monovalent Fab for binding in the future, while the whole IgG is essentially functional with bivalent potential.

The footprints of 7D6 and 6D6, from Fig. 3, appear to overlap the epitope of G32R7, an RBD-1 ("class 1" in the current authors' nomenclature) antibody and to border the epitope of S309 (the IgGs would probably compete). Thus, the authors' assignment of these two antibodies to this category is correct.

Response: We agree on the reviewer's suggestion to consolidate our 7D6/6D6 with S309 as single category in his/her first comments, which is more accurate with supports of overlapping amino acid site.